# β_2_-Integrin Adhesive Bond Tension under Shear Stress Modulates Cytosolic Calcium Flux and Neutrophil Inflammatory Response

**DOI:** 10.3390/cells11182822

**Published:** 2022-09-09

**Authors:** Vasilios Aris Morikis, Szu Jung Chen, Julianna Madigan, Myung Hyun Jo, Lisette Caroline Werba, Taekjip Ha, Scott Irwin Simon

**Affiliations:** 1Department of Biomedical Engineering, University of California-Davis, Davis, CA 95616, USA; 2Department of Biophysics & Biophysical Chemistry, Baltimore, MD 21218, USA; 3Biomedical Engineering, Johns Hopkins University, Baltimore, MD 21218, USA

**Keywords:** neutrophils, tension gauge tethers, β_2_-integrin

## Abstract

On arrested neutrophils a focal adhesive cluster of ~200 high affinity (HA) β_2_-integrin bonds under tension is sufficient to trigger Ca^2+^ flux that signals an increase in activation in direct proportion to increments in shear stress. We reasoned that a threshold tension acting on individual β_2_-integrin bonds provides a mechanical means of transducing the magnitude of fluid drag force into signals that enhance the efficiency of neutrophil recruitment and effector function. Tension gauge tethers (TGT) are a duplex of DNA nucleotides that rupture at a precise shear force, which increases with the extent of nucleotide overlap, ranging from a tolerance of 54pN to 12pN. TGT annealed to a substrate captures neutrophils via allosteric antibodies that stabilize LFA-1 in a high- or low-affinity conformation. Neutrophils sheared on TGT substrates were recorded in real time to form HA β_2_-integrin bonds and flux cytosolic Ca^2+^, which elicited shape change and downstream production of reactive oxygen species. A threshold force of 33pN triggered consolidation of HA β_2_-integrin bonds and triggered membrane influx of Ca^2+^, whereas an optimum tension of 54pN efficiently transduced activation at a level equivalent to chemotactic stimulation on ICAM-1. We conclude that neutrophils sense the level of fluid drag transduced through individual β_2_-integrin bonds, providing an intrinsic means to modulate inflammatory response in the microcirculation.

## 1. Introduction

Neutrophils are essential innate immune sentinels that are recruited into inflamed tissue in numbers to meet the host’s demand to control infection, ischemia, and necrosis [1]. They are endowed with an intrinsic capacity to overcome the fluid drag forces within the circulation through the activation of β_2_-integrin LFA-1 (CD11a/CD18) to a high-affinity (HA) state, which binds to ICAM-1 to ensure cell deceleration and shear-resistant arrest [2,3]. At the site of focal complexes supporting arrest, HA LFA-1 bond clusters trigger, from the outside in, a local influx of Ca^2+^ through linkage to the calcium release-activated channel (CRAC) Orai-1 [4]. This local increase activates a steady elevation of cytosolic Ca^2+^ through cooperation with endoplasmic reticular stores to reach a peak level necessary to synchronize the transition from cell arrest to shape polarization and migration [5,6,7]. Transmigration into peripheral tissue is signaled by the ligation of chemotactic GPCRs that superpose with integrin signaling to elicit maximum neutrophil activation that includes reactive oxygen species (ROS) production at levels sufficient to combat infection within parenchymal tissue [8]. Because the magnitude of tension acting on individual LFA-1/ICAM-1 bonds can regulate the level of cytoskeletal contractile response in T cell receptor activation [9], we reasoned that neutrophils similarly sense the magnitude of shear stress and convert this to outside-in signals that synchronize downstream effector functions.

A rapid burst of cytosolic Ca^2+^ triggers the transition from a symmetric spherical resting state characteristic of neutrophils in the circulation to a migratory polarized state endowed with an actin-rich pseudopod and myosin-rich uropod [10]. Neutrophils have the capacity to translate integrin receptor bond tension into an increase in cytosolic Ca^2+^ within a focal adhesion complex containing as few as 200 high-affinity (HA) LFA-1/ICAM-1 bonds. This is mediated by linkage with the cytosolic adaptor protein kindlin-3 and subsequent dissociation of RACK1 that triggers Ca^2+^ entry through Orai-1 [5]. Above a threshold level of tension exerted on a cluster of HA LFA-1 bonds, we observed that the rate and extent of Ca^2+^ flux increased in proportion to the shear stress applied over a narrow range from 0.5–2.0 dynes/cm^2^ [5]. Others have also reported that the dynamics of integrin clustering increases depending on the stiffness of the substrate and the magnitude of force applied to the cell membrane, implying that a feedback loop exists between integrin engagement, receptor clustering, and downstream Ca^2+^ signaling [11,12]. This raises a central question: is the efficiency of neutrophil recruitment to inflamed microcirculation throughout the body modulated by the relative shear stress, which varies with vascular structure and geometry? In the current study, we address the mechanism by which neutrophils transduce and thereby sense the magnitude of force transmitted from the outside in via LFA-1 to regulate downstream effector functions by modulating cytosolic Ca^2+^ flux.

Vascular mimetic microfluidic channels are used to simulate the flow in postcapillary venules and provide a means to capture neutrophils via LFA-1 on a glass substrate derivatized with recombinant ICAM-1, allowing observation of outside-in integrin-mediated signaling [13]. It is established that tensile force must be transmitted across the HA conformation of LFA-1 to trigger local CRAC Ca^2+^ entry that consolidates the focal adhesions required to overcome hydrodynamic drag force [14,15]. We and others have reported that dimeric HA LFA-1 bound to ICAM-1 behaves like a catch-bond in which the strength and lifetime increase with applied force up to bond rupture over a narrow range of shear stress [16,17]. Thus, integrin–ligand interactions generate traction forces that produce gradients in cytosolic Ca^2+^, which in turn modulate neutrophil arrest and subsequent diapedesis. In this manner, a burst in cytosolic Ca^2+^ through cooperation of LFA-1/kindlin-3/Orai1 and ER stores promotes ATP synthesis via the Krebs cycle and oxidative phosphorylation in the mitochondria. This in turn amplifies the cell’s metabolic rate to boost downstream ROS production [18]. Therefore, a deeper understanding is required of how the magnitude of fluid shear force acting on LFA-1 bonds in the circulation regulates the neutrophil adhesion cascade and subsequent inflammatory response within inflamed tissue.

In the current study, we endeavored to measure the relationship between tension acting on individual HA LFA-1 bonds and outside-in signaling by utilizing tension gauge tethers (TGT). TGTs are DNA duplex constructs designed to rupture above a threshold force and have recently been employed to study forces on individual LFA-1/ICAM-1 bonds during antigen recognition [9]. Here, we employed TGT that rupture at 12pN, 33pN, or 54pN that were biotinylated at one end of the nucleotide duplex and annealed to neutravidin-coated substrates on the floor of microfluidic channels. On the anterior side of the TGT, allosteric antibodies were assembled on protein-A/G to capture neutrophils through binding to β_2_-integrin heterodimers stabilized at low-affinity (TS1/18) or high-affinity (CBRLFA1/2) conformations. This system enabled direct observation of neutrophil response under shear stress supported by individual β_2_-integrin bonds that are loaded up to the prescribed rupture tolerance. It is important to note that we cannot distinguish between interactions involving the capture of one or two heterodimers by the divalent allosteric anti-CD18/TGT complex. Yet, evidence from FRET and single-molecule mechanics experiments implies that most attachments of HA β_2_-integrin are divalent connections to a clustered pair of heterodimers [19,20]. We discovered that a threshold force of 33pN across HA β_2_-integrin was required to trigger Ca^2+^ flux, whereas an optimum tension of 54pN efficiently elicited synergistic coupling between membrane CRAC influx and a burst of ER-mediated cytosolic Ca^2+^ at levels sufficient to amplify downstream ROS production.

## 2. Materials and Methods

Recombinant ICAM-1-Fc was purchased from R&D Systems (Minneapolis, MN, United States). Protein A/G was purchased from Fisher Scientific (Pittsburgh, PA, United States). Vybrant Cell-Labeling solutions were purchased from Life Technologies (Grand Island, NY, United States). Fluorescent conjugations and purified versions of anti-human CD18 monoclonal antibodies that identify HA β_2_-integrin (mAb24, binds to the MIDAS domain), affinity-modifying antibodies that target β_2_-integrin (CBRLFA1/2, which activates CD18 in a HA conformation, as does 240Q (provided by Ely Lilly, Indianapolis)), TS1/18 (which locks CD18 in an LA conformation) [21], anti-CD11b antibody that blocks Mac-1 function (M1/70), anti-CD45 antibodies (2D1), and fixation buffer and permeabilization buffer were purchased from Biolegend (San Diego, CA, United States). Function blocking anti-CD18 (IB4) was purchased from Santa Cruz Biotechnology (Santa Cruz, CA, United States). TruStain FcX™ and fixation buffer were purchased from Biolegend (San Diego, CA, United States). Antibodies were used at a saturating concentration of 5–10 µg/mL or per manufacturer’s instruction. Neutrophil-like HL-60 cells were purchased from ATCC along with DMSO, FBS, RPMI 1640, and anti/anti Pen/Strep (Manassas, VA, United States). N-formyl-Met-Leu-Phe (fMLP) and Phorbol 12-myristate 13-acetate were purchased from Sigma-Aldrich (St. Louis, MO, United States). Calcium indicators Rhod-2 AM (ex/em of bound Calcium 552/581) and Fura-2 AM (ratiometric dye excitation ratio 340/380), Poly-L-lysine solution, and dihydrorhodamine 123 were purchased from Thermo Fisher Scientific (Waltham, MA, United States). Avidin coated beads (d = 1μm) were purchased from Polysciences Inc. (Warrington, PA, United States).

### 2.1. PMN Isolation

Neutrophils were isolated from freshly collected human blood from consenting healthy donors according to protocol #235586–9 approved by The University of California, Davis Institutional Review Board. In some cases, whole blood was layered over the neutrophil separation medium Polymorphoprep (Cosmo Bio, Carlsbad, CA, USA) in a 1:1 ratio. After centrifugation, neutrophil cell layers were extracted and washed with HBSS containing 0.1% human serum albumin prior to use in experimentation. In other cases, neutrophils were isolated from whole blood via negative enrichment using an EasySep direct isolation kit purchased from StemCell Technologies, as per the manufacturer’s instruction (Cambridge, MA, United States). In brief, whole blood was incubated 1:1 with PBS containing the neutrophil isolation cocktail and magnetic RapidSpheres and incubated for 5 min. Cells were then placed in the EasySep magnet for 5 min. Cells were poured into a new tube and treated for an additional 5 min with 100 µL of magnetic RapidSpheres and placed on the EasySep magnet for 5 min twice more. Cells were then spun down and resuspended in HBSS containing 0.1% human serum albumin prior to use in experimentation.

### 2.2. HL60 Preparation

HL-60 cells were maintained in Advanced RPMI containing 10% FBS and 1% pen/strep. HL-60 cells were washed once a day for a minimum of three days to ensure optimal growth. These HL-60s were transfected with control scrambled siRNA or kindlin-3-specific siRNA (Santa Cruz Biotechnology, Santa Cruz, CA, USA) by electroporation using the Amaxa Nucleofector 4D using the SF Cell line 4D-nucleofector X Kit L (Lonza, Basel, Switzerland), according to the manufacturer’s instructions and as described previously [4,5]. HL-60 cells were given fresh media 24 h before electroporation to maximize survival. Cells were pelleted from the media, resuspended in high-resistance nucleofection buffer in the presence of 100 nM siRNA, and electroporated. Immediately after electroporation, cells were transferred to 37 °C media containing 1.3% DMSO and differentiated for 3–5 days under the influence of siRNA prior to experimentation [4,5]. It was confirmed by Western blot that kindlin-3 expression was silenced up to 90% in dHL-60 cells. Kindlin-3 siRNA-scrambled mutation R556A was used as a control.

### 2.3. Microfluidic Design and Fabrication

Microfluidic devices were designed to have four independent flow channels to analyze multiple conditions per coverslip. The channels have dimensions of 60 µm × 2 mm × 8 mm (h × w × l) and are surrounded by a spiderweb of vacuum channels to allow for a reversible sealing to functionalized glass coverslips. The microfluidic devices were designed in AutoCAD (Autodesk, San Rafael, CA, United States), and then a printed negative photomask was made by CAD/Art Services Inc. (Bandon, OR, United States). To create master molds, SU-8 50 photoresist (MicroChem, Newton, MA, United States) was spun down to a height of 60 µm on a 200 mm diameter silicon wafer. The photomask was placed on top of the wafer, and UV light was shone through to crosslink the areas of interest as printed on the photomask. Sylgard 184 elastomer and curing agent (Dow Corning, Midland, MI, United States) were mixed 10:1 and cast over the negative master and then baked to create PDMS replicas. Flow and vacuum access holes were then punched into the PDMS devices using a blunted needle.

### 2.4. TGT Preparation, Glass Coverslip Functionalization, and Device Assembly

Tension gauge tethers (TGT) with maximum force thresholds of 12pN, 33pN, and 54pN at 20µM were prepared as described previously [22,23]. Briefly, the dsDNA serves as a rupturable tether with a tension tolerance dependent upon the position of the linkage to a biotin chemically linked at a predetermined position. This is tethered through an avidin–biotin bond. Antibodies—CBRLFA1/2, TS1/18, 2D1, or ICAM-1—were incubated at 500 nM at 4 °C for 1 h with TGT-Protein-A/G constructs on the outward strand of the dsDNA. Microfluidic channels were assembled on PEGylated neutravidin substrates treated for 10 min with 20 nM TGT constructs. The surface was washed with PBS prior to cell perfusion. Human neutrophils were suspended in HBSS with 0.1% HSA at 1 × 10^6^ cells/mL and in most experiments were treated with the Mac-1 blocking antibody M1/70 prior to perfusion to block non-specific adhesion. In cases where calcium was measured, cells were incubated with 1 µM Rhod-2 AM or Fura-2 AM for 20 min at room temperature in the dark. Cells were spun down and resuspended and allowed to de-esterify for 15 min prior to use. Neutrophils were loaded into a reservoir and drawn through the system via a syringe pump (Cellix Ltd., Dublin, Ireland or Harvard Apparatus, Holliston, MA, United States) and allowed to adhere prior to shear ramping at appropriate flow rates, resulting in wall shear stresses between 0.1 dynes/cm^2^ and 8 dynes/cm^2^ at the glass–fluid interface. For the bead binding assays, avidin-coated beads were derivatized with TGT in the same manner as above at a surface density of ~21.6 × 10^3^ TGT/μm^2^. The beads were then washed three times with a solution of HBSS + 1% HSA and then blocked with 200 μg/mL fibrinogen for 4 h at 4 °C. The microfluidic channels described previously were plasma-bound to a non-functionalized glass surface and incubated with 100 μg/mL of fibrinogen for 2 h at room temperature. The beads were slowly perfused into the channels and allowed to settle overnight to promote firm adhesion. The bead-coated flow chambers were then washed with HBSS + 2% HSA for 10 min prior to the introduction of neutrophils.

Cells were imaged using 20× phase objective images taken every second for 5 min using a 16-bit digital complementary metal oxide semiconductor (CMOS) camera (Andor ZYLA) connected to a PC (Dell) with NIS Elements imaging software. Neutrophil arrest, calcium flux, and CD18 integrin density were all quantified. For neutrophil calcium flux quantification, neutrophils were observed pivoting over the bead during low flow, and calcium was measured after ramping. If the cells did not pivot or detached within 30 s of ramping the shear stress to 1 dyne/cm^2^, quantification was not performed. To measure cluster density an inverted TIRF research microscope (Nikon) equipped with a 60× numerical aperture 1.5 immersion TIRF objective, a motorized stage, and 488 nm and 543 nm solid state lasers as TIRF excitation light sources with the appropriate filter set was used. Neutrophils were sheared over TGT substrates in the presence of high-affinity reporting CD18 antibody (mAb24), and images were captured at regular intervals. To identify cluster density from MFI, the same setup was used to generate a standard curve. QSC beads with four distinct antibody binding site populations were incubated with AF488 mAb24 for 10 min, washed three times, and then injected into our microfluidic devices and allowed to settle for 5 min. Fluorescent intensity was observed using TIRF microscopy. The MFI of the observable surface area was correlated with the known number of adhesion sites, and a conversion equation was identified using ImageJ software (Appendix A).

### 2.5. ROS Production Measured in Neutrophils in a 96 Well Plate Reader

Clear polystyrene 96 well plates were precoated with recombinant ICAM-1 with poly-L-lysine solution. A final concentration of ICAM-1 (10 μg/mL) made from a 400 μg/mL stock solution was aliquoted and kept in a −80 °C freezer. Plates were washed three times with PBS then blocked with HBSS^+/+^ buffer + 0.1% HSA before being incubated for one hour at 37°C. HL-60 differentiated to neutrophils or neutrophils isolated from venipuncture collected blood was resuspended in HBSS^-/-^ buffer + 0.1% HSA and pretreated with Human TruStain FcX™. Purified anti-mouse/human anti-CD11b mAb or anti-CD18 mAb as indicated was used to treat neutrophils for 15 min before being washed and spun down at 400 rpm at 37 °C. Each well received 2 × 10^5^ cells to which 56 μM of dihydrorhodamine 123 buffer was added (Catalog: D23806, Thermo Fisher Scientific) with HBSS^+/+^ buffer + 0.1% HSA. Neutrophils were stimulated with/without 10 ng/mL of Phorbol 12-myristate 13-acetate, 10μg/mL of anti-CD18 240Q mAb, or 1μM of fMLP to activate HA CD18. To simulate shear, cells were mixed every 2.5 min. The ROS signal was measured using a Synergy™ HT Multi-Mode Microplate Reader over 150 min.

To examine the role of Ca^2+^ in ROS production, freshly isolated neutrophils were resuspended in HBSS-/- buffer + 0.1% HSA and then blocked with Human TruStain FcX™, purified anti-CD11b antibody, and thapsigargin with/without 2-APB for 15 min, 400 rpm at 37 °C. We added 2 × 10^5^ neutrophils to each well and then added 55.56 μM of dihydrorhodamine 123 buffer with HBSS+/+ buffer + 0.1% HSA. The ROS signal was measured using a Synergy™ HT Multi-Mode Microplate Reader.

To determine the dependence of ROS production on tension acting on β_2_-integrin, freshly isolated neutrophils were resuspended in HBSS^+/+^ buffer + 0.1% HSA with HA inducing anti-CD18 (240Q) and where indicated blocked with anti-CD18 (IB4) antibody for 15 min at room temperature as indicated. Neutrophils were incubated with 12-pn TGT or 54-pn TGT protein prebound with ICAM-1 for 30 min, rotated, at room temperature. Biotinylated TGT-ICAM-1 were annealed to the bottom of neutravidin-coated 96-well plates (Catalog: 15117, Thermo Fisher Scientific) and then incubated with 2 × 10^5^ neutrophils bound and allowed to adhere for 10 min before the addition of 55.56 μM of dihydrorhodamine 123 buffer with or without fMLP in HBSS^+/+^ buffer + 0.1% HSA. To simulate shear, cells were mixed every 2 min. The ROS signal was measured using a Synergy™ HT Multi-Mode Microplate Reader.

## 3. Results

### 3.1. β2-Integrin Bond Tension over a Narrow Range Modulates Neutrophil Adhesion Strength

Neutrophil arrest on inflamed endothelium involves a precise balance between the drag force of fluid shear stress (τ) that is resisted by tension acting on HA β_2_-integrin bound to ICAM-1, which consolidates into bond clusters that facilitate stable cell arrest on inflamed endothelium (Figure 1a). Previously, we reported that the level of cytosolic Ca^2+^ flux measured on neutrophils anchored via ~200 HA LFA-1 bonds within a single site of focal complexes increased in proportion to the magnitude of the fluid shear stress [5]. Here, we delve into how bond tension transmitted across a single HA β_2_-integrin bond modulates cytosolic Ca^2+^ flux, which results in the consolidation of bond clusters and downstream signaling of effector functions on individual neutrophils. Neutrophils were imaged upon capture by anti-CD18 allosteric inducing antibodies that upon binding to the integrin β_2_-subunit stabilize an HA (240Q or CBRLFA-1/2) or LA (TS1/18) conformation, respectively [24]. We have previously reported that neutrophils mechanosignal Ca^2+^ flux and β_2_-integrin clustering when bound to ICAM-1 under fluid shear stress that is equivalent to the capture of CD18 via HA-inducing allosteric antibodies [4,5]. In this study, we assessed the magnitude of force necessary to transduce signaling in neutrophils bound to allosteric antibodies annealed to the flow channel substrate through a TGT–DNA duplex that ruptures at a prescribed tolerance when sheared by antiparallel forces [23]. A threshold force of 54pN will rupture a TGT when the biotin anchor is positioned 18 base pairs from the opposite end of the duplex where a protein-A/G bound anti-β_2_-integrin antibody is positioned (Figure 1a). Positioning the biotin anchor at an intermediate position along the duplex results in a threshold rupture tension of 33pN, whereas at the lowest threshold of 12pN, the biotin is positioned opposite the anti-β_2_-integrin allosteric antibody. In this manner, we performed separate microfluidic experiments on substrates coated with TGTs that rupture over a range of 54pN, 33pN, and 12pN of tension per bond. Neutrophils were perfused through the microfluidic channels at an initial shear stress of 0.25 dynes/cm^2^ to allow cells to settle and bind to the anti-β_2_-integrin/TGT construct before the flow rate was incrementally ramped up to the indicated shear stress for 2 min prior to quantitation of the number of neutrophils/FOV that remained bound at each shear stress and for each TGT tolerance (Figure 1b). At a shear stress of 0.25 dyne/cm^2^ the theoretical maximum tension acting on a spherical neutrophil held by a tether is ~7pN/bond [14]. Bond tension is predicted to increase linearly with fluid drag force up to 104pN/bond at 2 dyne/cm^2^. Neutrophil detachment commenced at 1 dyne/cm^2^ for the 33pN and 12pN TGT-bound cells, captured on HA-inducing allosteric mAbs in relative proportion to the respective TGT threshold tolerance (Figure 1b). In contrast, nearly all neutrophils captured via HA β_2_-integrin presented on 54pN TGT, or directly to the substrate (denoted CBR LFA1/2 in Figure 1b), remained arrested up to 8 dynes/cm^2^. Neutrophils that remained bound to 12pN TGT with increased shear stress did not effectively increase the cluster density of HA β_2_-integrin bonds, whereas those bound to 33pN and 54pN TGTs effectively consolidated bond density up to a maximum of ~5000 HA β_2_-integrin receptors per cluster (Figure 1c,d,f). This maximum of β_2_-integrin cluster density with shear ramp above 1 dyne/cm^2^ was only maintained in neutrophils captured via HA β_2_-integrin receptors on 54pN TGT or CBRLFA-1/2 directly annealed to the substrate, whereas cells bound via LA β_2_-integrin (TS1/18) directly annealed to the substrate remained attached only up to 4 dyne/cm^2^ before dissociation. These data reveal the importance of tension on HA β_2_-integrin in the consolidation of bond clusters necessary for shear strengthening with increased fluid drag force that is achieved at 54pN, but not 33pN or 12pN TGTs presenting CBR LFA1/2 or neutrophils captured at LA with TS1/18. Noteworthy is the capacity for nearly all neutrophils bound via 54pN TGT to rapidly undergo shape change from a high aspect ratio spherical to a flattened low-profile shape (Figure 1e,f). Neutrophil shape change was defined as a 50% increase in surface area as detected by the proportional increase in area of Rhod-2 positive cells. Less than 50% of neutrophils attached to 33pN TGT and less than 10% of those attached to 12pN TGT exhibited shape change, and those that did flattened more slowly. By comparison, HA β_2_-integrin activated on neutrophils by the addition of 240Q to the perfusion buffer and captured on a substrate of ICAM-1 produced the most shear-resistant neutrophils supported by diffusive HA β_2_-integrin bonds that consolidated into large clusters, which rapidly transitioned into a polarized shape characterized by the formation of a uropod and pseudopod, as previously reported [3]. Few neutrophils captured via LA β_2_-integrin on TGT presenting TS1/18 antibody exhibited shape change at any shear stress, nor did they consolidate HA β_2_-integrin and exhibit shear strengthening (data not shown). We conclude that the application of bond tension between 33pN and 54pN acting on HA β_2_-integrin results in more efficient focal adhesive cluster formation that supports shear-resistant cell arrest and conversion to a migratory state.

### 3.2. A Narrow Range of β_2_-Integrin Bond Tension Triggers Efficient Ca^2+^ Flux in Arrested Neutrophils

Tension on HA β_2_-integrin induces transmembrane binding of kindlin-3 to the β-subunit cytodomain, which in turn forms a complex with Orai1 to elicit a burst of cytosolic calcium that promotes shear strengthening and neutrophil polarization [4,5]. Moreover, we have reported that the dynamics of HA β_2_-integrin bond cluster formation rise in concert with Ca^2+^ flux as the magnitude of the shear stress acting on arrested neutrophils is incrementally increased [5]. Here, we examined the dynamics of Ca^2+^ flux as a function of β_2_-integrin bond tension by anchoring neutrophils on each TGT substrate at a constant shear stress of 1 dyne/cm^2^. The dynamics of cytosolic Ca^2+^ release were quantified for cells captured on TGTs presenting CBR LFA-1/2 or TS1/18. Neutrophils captured via HA β_2_-integrin on 54pN TGT exhibited a continuous, rapid rise in cytosolic Ca^2+^, which consistently reached a peak level at roughly sixfold above baseline within 30 s of the ramp in shear stress (Figure 2a,b). In contrast, neutrophils sheared on 54pN TGT presenting TS1/18 to stabilize LA β_2_-integrin did not register a rise in Ca^2+^ above baseline (data not shown). Remarkably, neutrophils captured via HA β_2_-integrin on 33pN TGT exhibited very different dynamics characterized by inconsistent sparks of cytosolic Ca^2+^ that rose more slowly above baseline (Figure 2c,d and Appendix A). Comparing the frequency distribution of Ca^2+^ flux for neutrophils sheared on each TGT revealed a normal distribution centered at sixfold above baseline for the 54pN TGT. In contrast, neutrophils attached via 33pN TGT exhibited a bimodal distribution, rising slowly to a twofold increase, while those on 12pN remained at the baseline level. Taken together, these data reveal that bond tension above a threshold of 33pN determines the efficiency of ignition and subsequent strength of Ca^2+^ signaling.

Previously, we reported that neutrophils sheared in a microfluidic channel presenting beads coated with CBRLFA-1/2 activated ~200 HA β_2_-integrin bonds per cell. These bonds were loaded by ramping up the flow rate, causing the cell to pivot downstream of the bead that triggered a rapid rise in Ca^2+^ flux that peaked at a level proportional to the magnitude of the fluid shear stress [5]. Here, we assembled a similar system by annealing 54pN or 33pN TGT to beads and examining how bond tension at a prescribed HA β_2_-integrin density influences Ca^2+^ flux (Figure 3a). Neutrophils were perfused into the microfluidic channel at a low flow rate, allowing stable capture on beads, and the shear stress was ramped up to 0.5 dynes/cm^2^. Upon pivoting over the bead and exerting tension on β_2_-integrin supported by TGT-anti-β_2_-integrin, neutrophils exhibited a typical fivefold rise in Ca^2+^ flux that was significantly more rapid and reached a higher peak level when attached via 54pN compared with 33pN TGT (Figure 3b). These data indicate that the rate and extent of Ca^2+^ influx is amplified above a threshold in bond tension acting on an equivalent number of HA β_2_-integrin bonds.

### 3.3. Reactive Oxygen Species Production Is Amplified by Mechanosignaling via HA LFA-1

Chemotactic signaling of neutrophils stimulated via GPCR by the addition of fMLP signals the release of PLC, which results in cytosolic Ca^2+^ release by ER calciosomes [25]. To amplify downstream production of reactive oxygen species at the site of inflammation, adherent neutrophils receive a second signal via β_2_-integrin bonds [26,27]. We assessed the role of HA LFA-1 conformation and bond tension in the regulation of the dynamics of ROS production by allowing neutrophils to adhere to the bottom of 96 well plates coated with ICAM-1 in the presence of the allosteric-inducing antibody 240Q following pretreatment with anti-Mac-1 (Figure 4a). Neutrophil activation by addition of the calcium ionophore PMA elicited the fastest rate and highest extent of ROS production. Allosteric induction of HA β_2_-integrin in the presence of 240Q, or stimulation via fMLP, mediated adhesion to ICAM-1 and an equivalent level of ROS production that reached ~50% of that activated with PMA stimulation. The superposition of fMLP and 240Q, but not an isotype control IgG (Appendix A), engendered a significant increase in ROS production of up to ~75% of the maximum elicited by PMA (Figure 4b,c). The mechanism by which ROS production is amplified was attributed to an increase in LFA-1-dependent shear-resistant adhesion in the presence of blocking anti-Mac-1 mAb. Induction of HA β_2_-integrin with 240Q supported a significant increase in firmly attached neutrophils compared to fMLP or PMA (Figure 4d). In contrast, ROS production was reduced to baseline when captured by TS1/18 (data not shown) to stabilize LA β_2_-integrin or by pretreatment with IB4 to block CD18 adhesion to ICAM-1 (Figure 4c). Furthermore, shear mixing of neutrophils adherent to the well plates was necessary to generate tension on HA β_2_-integrin and amplification of the rate and extent of ROS production, which was not observed under static conditions (Appendix A).

### 3.4. Mechanosensitive Reactive Oxygen Species Production Is Regulated by Kindlin-3

To dissect the mechanism by which tension acting on HA β_2_-integrin bonds amplifies ROS production, neutrophils were pretreated with thapsigargin (Tg) in Ca^2+^-free media to deplete cytosolic stores and activate Ca^2+^ influx and captured on the well bottom by allosteric mAbs (Figure 5a). Previously, we established that adding back a bolus of buffer containing 1.5 mM CaCl_2_ effectively served as an agonist that stimulates the opening of Orai1 CRAC channels, which in turn elicits downstream neutrophil arrest and shape change [13]. Inducing SOCE-activated neutrophil adhesion on allosteric HA-inducing CBRLFA-1/2 elicited a threefold increase in ROS production, equivalent to capture on ICAM-1 in the presence of 240Q (Figure 5b). In contrast, neutrophils captured on TS1/18, to stabilize LA β_2_-integrin, registered a 30% reduction in ROS production. We next confirmed that the SOCE-mediated activation of ROS production was dependent on membrane calcium channel influx by pretreating neutrophils with the CRAC channel antagonist 2-APB, which we have previously reported virtually eliminates Ca^2+^ entry via Orai1 (Figure 5a) [4]. Importantly, when bound to ICAM-1 alone and activated by Ca^2+^ repletion, neutrophils exhibit increased ROS production compared to those bound to the LA-inducing antibody TS1/18 (Figure 5B). Blocking CRAC channels with 2-APB lowered ROS fluorescence to the baseline level observed in untreated neutrophils in the absence of stimulus (Figure 5b). These data confirm that in Tg calcium-depleted neutrophils, Ca^2+^ influx mediated via membrane CRAC is sufficient to trigger HA β_2_-integrin binding to ICAM-1 that, under bond tension, signals downstream ROS production.

To confirm that integrin-mediated amplification of ROS production is transduced through HA β_2_-integrin that forms a complex with its cytoplasmic adaptor kindlin-3, we measured the kinetics of DHRhod in HL-60 myeloid cells differentiated into neutrophils (dHL60). We previously reported that dHL60 cells express neutrophil receptors including LFA-1 and Mac-1 that can be activated with fMLP or allosteric induction with 240Q [5]. Moreover, we demonstrated that HL60 transfected with siRNA to transiently silence kindlin-3 expression lacked the capacity to fluctuate Ca^2+^ via CRAC and activate downstream shear-resistant arrest and chemotaxis [13]. In the current study, we measured ROS production in kindlin-3 knockdown dHL60s compared with wild-type or scrambled siRNA control dHL60s to determine whether kindlin-3 is required for CRAC activation and downstream ROS production (Figure 5c). Neutrophil ROS production above the baseline level was observed with stimulation via fMLP (~1.5 fold increase) or 240Q in suspension (~1.2-fold increase), but the combination of allosteric and chemotactic stimulation increased ROS production by ~3.5-fold over untreated dHL60s (Figure 5d). Silencing kindlin-3 in dHL60s suppressed the amplification in ROS production elicited cooperatively by fMLP and 240Q stimulation, similar to the effect of blocking Orai1 CRAC activity with 2-APB. It is important to note that silencing kindlin-3 in dHL60 cells did not impair the outside-in allosteric activation by 240Q of ICAM-1 adhesion compared to scrambled control siRNA dHL60s (Appendix A).

### 3.5. A Threshold β_2_-Integrin Bond Tension Amplifies ROS Production in Adherent Neutrophils

Finally, we assessed whether tension transduced via HA β_2_-integrin effectively modulates the extent of ROS production in neutrophils captured on a substrate coated with TGT-ICAM-1 and stimulated by fMLP (Figure 6a). Neutrophils adherent to TGT-ICAM-1 annealed to the substrate registered equivalent baseline levels of ROS that are not significantly greater than in controls attached via an IgG isotype (Figure 6b and Appendix A). Following fMLP stimulation, 54pN TGT-captured neutrophils produced significantly more ROS than those attached to 12 pN TGT (Figure 6b). This was evident in samples where neutrophil adhesion via CD18 was blocked by pretreatment with IB4, which registered a significant drop in ROS production for adhesion to ICAM-1 presented on 54pN but not 12pN TGT. We conclude that tension acting on neutrophils firmly adherent to 54pN TGT-ICAM-1 elicited a 50% increase in ROS production due to cooperative signaling between HA β_2_-integrin and fMLP, which was not detected in cells captured via 12pN TGT.

## 4. Discussion

A wealth of published data supports the contention that neutrophil arrest mediated by β_2_-integrin bonds under tension transmits outside-in signals that synchronize the sequential steps necessary for neutrophils to navigate to appropriate vascular sites of tissue inflammation. Here, we report that neutrophil capture on allosteric antibodies linked to TGT with sufficient strength promotes neutrophil adhesion via HA β_2_-integrin bonds that mechanosignal Ca^2+^ flux. A threshold level of tension above 33pN acting on individual β_2_-integrin receptors within a focal cluster was necessary and sufficient to efficiently generate a burst in cytosolic Ca^2+^ via cooperation between ER stores and a cytosolic complex that includes kindlin-3 and membrane CRAC. Neutrophils exposed to low shear stress on the order of 0.25 dyne/cm^2^, or at higher shear stress but weakly tethered via 12pN TGT, maintained a spherical shape that correlated with low levels of cytosolic Ca^2+^, commensurate with the resting state in the circulation. In contrast, neutrophils tethered via 54pN TGT remained firmly adherent to the substrate, supported by increased consolidation of HA β_2_-integrin bond clusters that preceded a rapid rise in cytosolic Ca^2+^ and shape polarization, characteristic of cell arrest on ICAM-1 at sites of inflammation. Ca^2+^ flux was observed in only 40% of neutrophils captured via 33pN TGT, which exhibited a markedly slower rise and lower peak level than those attached to 54pN TGT. This correlated with diminished capacity to consolidate HA β_2_-integrin bonds into the clusters necessary to support shear-resistant adhesion. These data provide clear evidence that tension sensed over a narrow range by HA β_2_-integrin and transmitted across focal complexes modulates secondary messenger releases that tune the rate and extent of neutrophil responses, including migratory and secretory functions.

A confounding factor in comparing neutrophils tethered to the various TGTs presented on the flat glass substrate in the microfluidic channels is that the number and the density of HA β_2_-integrin clusters varied greatly for 54pN versus 12pN and 33pN TGTs at stresses >1 dynes/cm^2^. However, the capture of neutrophils on TGT-coated beads at equivalent contact areas and bond densities revealed a threshold response in the modulation of Ca^2+^ flux of neutrophils tethered via 33pN versus 54pN TGT. The dynamics in the rise of the Ca^2+^ signal for neutrophils pivoting over TGT-coated beads were slower and the frequency distribution more uniform than for neutrophils sheared in the flat substrate of the microfluidic channels (i.e., Figure 2b versus Figure 3b). Neutrophils tethered on 54pN TGT exhibited a more rapid rise to a fivefold peak above baseline as compared with a twofold peak for bonds under 33pN of tension. We envision that HA β_2_-integrin bonds form rapidly upon contact with TGTs and transmit tension to cytosolic adaptors such as kindlin-3 that anchor the receptor in the membrane. As tension rises, we expect that a greater number of HA β_2_-integrins transmit force up to the respective TGT threshold. We envision that bonds under 54pN more efficiently engage kindlin-3 at the cytodomain of HA β_2_-integrin. This in turn triggers a feed-forward signaling mechanism that involves the consolidation of HA bonds, the recruitment of Orai1 CRAC channels, and Ca^2+^ influx. While HA β_2_-integrin bonds tethered to 33pN TGT were sufficient to maintain adhesive contact to the beads, this tension was insufficient to consistently trigger consolidation of HA bond clusters and local Ca^2+^ flux within focal adhesive contacts.

In the current study, LFA-1 was bound by allosteric activating or inhibiting antibodies linked to TGT such that force below the rupture tension could engage one or two LFA-1 heterodimers at each Fab of the antibody. In the case of dimeric bonds, the two LFA-1 heterodimers would equally share the force; thus, a 54pN TGT could only transmit up to 27pN of tension to each integrin heterodimer before rupturing. Likewise, two LFA-1 heterodimers bound to a 33pN or 12pN TGT would experience at maximum ~16.5pN and 6pN of force, respectively. A continuum mechanical model of the forces acting on a spherical neutrophil arrested on inflamed endothelium in a postcapillary venule experiencing a shear stress ranging from 1–2 dynes/cm^2^ is estimated to be ~30–60pN per bond [28]. We observed that neutrophil activation was efficiently transduced when tethered via 54pN TGT, which for bivalent binding translates to ~27pN of tension/receptor. In contrast, we observed that for capture on a 33pN TGT and ~16.5pN/receptor, less than 50% of neutrophils fluxed Ca^2+^ and subsequently spread on the substrate. Thus, we believe tension exceeding ~16.5pN per bond is necessary to mechanically stabilize LFA-1/ICAM-1 and initiate linkage to kindlin-3, which promotes LFA-1 clustering and association with Orai1-mediated Ca^2+^ signaling. A similar magnitude of tension acting on LFA-1 was recently demonstrated to activate T cell cytoskeletal force generation and cell spreading on a substrate of TGT [9].

In considering the conformation of β_2_-integrin necessary to observe tensile force-dependent modulation in mechanosignaling, it is noteworthy that CBR LFA1/2 that binds the EGF-3 domain of β_2_-integrin and induces opening of the headpiece and spreading of the α and β subunits is sufficient for outside-in transduction of the signal (Schürpf & Springer, 2011). Moreover, we showed that mAb 240Q, which recognizes the I-domain allosteric site and induces HA β_2_-integrin to ICAM-1, could induce tension-mediated amplification in Ca^2+^ flux and ROS production in neutrophils sheared on ICAM-1/TGT complexes. In contrast, TS1/18 that recognizes the β-subunit I-domain and stabilizes LA β_2_-integrin is not sufficient to generate a signal, no matter the level of bond tension applied.

Dynamic force spectroscopy measurements of the strength and lifetime of LFA-1 binding to ICAM-1 over a wide range of loading rates has shown that bonds behave as slip bonds when the heterodimer is locked in a high-affinity state in the presence of activating divalent cations such as Mg^2+^ or allosteric-inducing antibody 240Q [29]. In the HA conformation, LFA-1 binds ICAM-1 with persistent mechanical strength (>20pN for >1 s) even when subjected to relatively slow increases in tension (<15 pN/s). Provided that LFA-1 is allosterically activated, the mechanical strength and bond lifetime is equivalent for monomeric and dimeric bonds to ICAM-1. The enhanced bond strength with increased tension is attributed to a shift in the α7 domain of the LFA-1 headpiece that engages ICAM-1 with high affinity. Only in this conformation can force be transmitted from outside transmembrane to the cytosolic tail [30].

Currently, it is not known how kindlin-3 may contribute to conversion of LFA-1 to a HA conformation, as tension-mediated association with talin, which binds and activates LFA-1 in response to GPCR signaling, is not reported. We surmise that a mechanical feed-forward process is initiated by a transmission of force exceeding 16.5pN to the β_2_-integrin cytoplasmic tail. However, it remains unknown how force catalyzes the durable binding of kindlin-3 and promotes linkage with neighboring LFA-1 heterodimers to consolidate HA bonds within focal sites of contact. One possibility, which is supported by structural analysis of kindlin-2 functioning in platelets, is that tension disrupts residues in the β subunit of HA LFA-1 to enable linkage to the F3 subdomain of kindlin-3. In the current study, we did not directly measure the level of kindlin-3/LFA-1 linkage as a function of TGT threshold tension. However, previously we reported that the magnitude of shear stress acting on HA LFA-1 correlates directly with the extent of association between LFA-1/kindlin-3/Orai1 as detected by Western blot. The PH domain of kindlin-3 is essential for integrin-dependent adhesion and binds with high affinity to PI(3,4,5)P3 and paxillin, which is necessary for focal adhesion assembly and the induction of actin-dependent migratory function. This may increase kindlin-3 dimerization and association with membrane PIP3 anchors and paxillin to promote its linkage to the actin cytoskeleton. Although the current study is focused on HA β_2_-integrin interacting with ICAM-1, it is worth noting that LFA-1 also recognizes ICAM-2 and ICAM-3, which are expressed on a variety of cells including lymphocytes, monocytes, neutrophils, and endothelial cells, and the role of bond tension in regulation of mechanosignaling between cells expressing these co-receptors is unknown [31].

Neutrophils activated by GPCRs such as fMLP are primed for amplified ROS production by cooperative mechanosignaling via HA LFA-1 binding to ICAM-1 [32]. The engagement of these receptors initiates intracellular signaling including phospholipase beta (PLCβ) and PI3K-dependent signaling that leads to IP3-mediated release of ER Ca^2+^ stores and assembly of the NADPH oxidase complex [33]. It is well published that fMLP stimulation of neutrophils to activate integrin mediated binding to fibrinogen induces ROS production [34,35,36,37,38]. Likewise, activation with allosteric antibodies is sufficient to elicit strong ROS production in neutrophils, revealing that β_2_-integrin itself can activate cells. We demonstrated that superposition of signaling via HA-LFA-1 and fMLP significantly amplified ROS production in neutrophils adherent via LFA-1 to ICAM-1 that required sufficient bond tension. Previously, we reported that Ca^2+^ influx induced by thapsigargin is signaled by HA LFA-1/kindlin-3/Orai1 complex. Here, we showed that amplification of ROS production by superposition between fMLP and HA LFA-1 required Ca^2+^ mechanotransduction via kindlin-3 that was inhibited by siRNA knockdown. Furthermore, we demonstrated that Ca^2+^ entry activated by LFA-1 bond tension was mediated by CRAC channel opening, as ROS production in thapsigargin-treated neutrophils was blocked in the presence of 2-APB. Finally, we showed that cooperative signaling between HA LFA-1 and fMLP-stimulated neutrophils, which fluctuate Ca^2+^ within minutes under shear stress, resulted in a 50% amplification of ROS production signaled by neutrophils’ capture on 54pN but not 12pN TGT-ICAM-1. We conclude that effector function elicited downstream of β_2_-integrin mechanosignaling within minutes of arrest under shear stress in circulation can modulate neutrophil ROS production over hours within inflamed tissue.

In summary, we demonstrate that a threshold force acting on HA β_2_-integrin is necessary to trigger mechanoregulation of Ca^2+^-mediated outside-in signaling. This provides a means of spatiotemporal control of neutrophil recruitment through modulation of the strength of adhesion through consolidation of a focal adhesion complex supported by HA β_2_-integrin bound to ICAM-1 on inflamed endothelium. At vascular sites of inflammation, low shear flow is associated with ischemia. We speculate that at sufficient flow rates and corresponding fluid drag acting on tethered neutrophils, tension acting on engaged integrins combines with GPCR signaling via chemokines to mechanoregulate the rate and extent of cytosolic Ca^2+^ flux. This in turn modulates the extent of neutrophil shape change from a spherical to a polarized cross-sectional profile that lessens the drag force acting and facilitates shear-strengthened cell arrest and migration. While the latter multistep process of capture–arrest–diapedesis that is mechanoregulated via cytosolic Ca^2+^ occurs over minutes, it can modulate effector functions within inflamed tissue, including the ROS production and NETosis necessary to combat pathogens and ensure tissue homeostasis that occur over hours to days.

## Figures and Tables

**Figure 1 cells-11-02822-f001:**
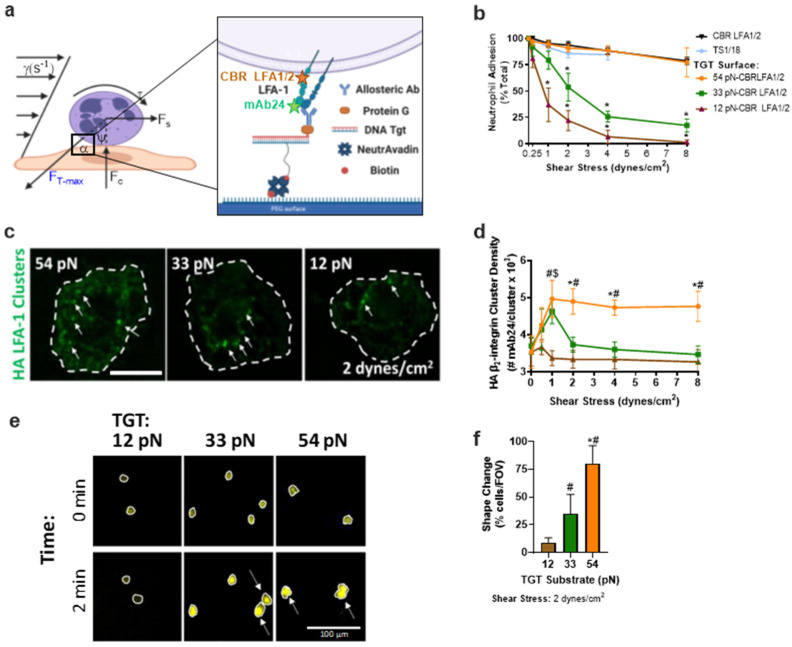
**Neutrophils’ adhesion strengthening depends upon the level of bond tension acting on HA****β****_2_****-****integrin.** (**a**) Schematic of neutrophils sheared in microfluidic channels bound to the glass coverslip via allosteric antibody, CBR LFA1/2 (binding epitope shown in orange) linked to one side of a TGT that is annealed to the substrate on glass coverslips. High-affinity β_2_-integrin was reported on using mAb24 (binding epitope shown by green star) (**b**) Neutrophils binding to each TGT bearing CBR LFA1/2 substrate quantified as the percentage of cells that remained bound after incrementally increasing shear stress, as defined on the x-axis. Arrested neutrophil percent is reported as mean +/− SEM over 5 fields of view with >10 cells per experiment (*n* = 3, * compared to CBR LFA1/2, *p* < 0.01). (**c**) AF488 mAb24 clusters (green) of bound neutrophils sheared over CBR LFA1/2:TGT substrate at (1 dynes/cm^2^) measured via TIRF; scale bar depicts 5 µm. (**d**) HA β_2_-integrin cluster density measured via TIRF. Graphs plot HA β_2_-integrin clustering mean +/− SEM for 5 fields of view for each experiment; scale bar shows 10 µm (*n* = 3, * 54pN and 33pN, # 54pN and 12pN, $ 33pN and 12pN, *p* < 0.01). (**e**) Neutrophils with Ca^2+^ reporter Rhod-2 sheared over TGT/HA-inducing β_2_-integrin substrate for 2 min at 2 dynes/cm^2^. White circles show area and arrows identify cells that have undergone shape change compared to time 0; scale bar shows 100 µm. (**f**) Quantification of the percentage of neutrophils that change shape after shearing for 2 min over TGT/HA-inducing β_2_-integrin substrate. Graphs plot percent shape change mean +/− SEM for 5 fields of view for each experiment (*n* = 3, * 54pN and 33pN, # 54pN/33pN and 12pN, *p* < 0.01, n.s.: not significance).

**Figure 2 cells-11-02822-f002:**
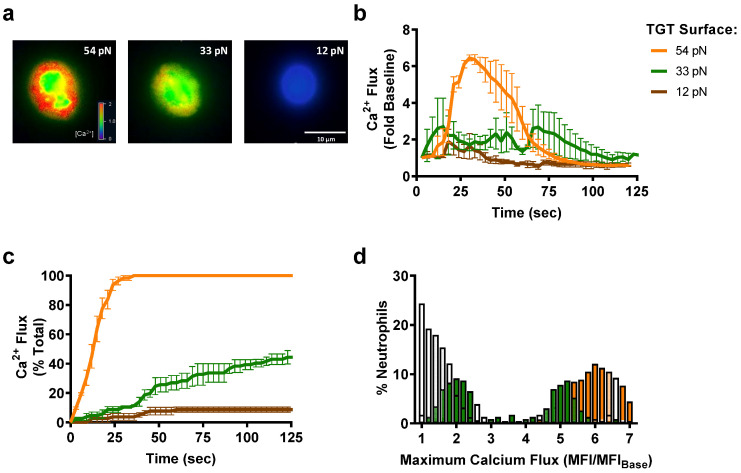
**Neutrophil Ca^2+^ flux amount and profile depend on the level of bond tension acting on HA β_2_-integrin.** (**a**) Representative images of neutrophils sheared over TGT substrates functionalized with HA-inducing β_2_-integrin (CBR LFA1/2). Ca^2+^ quantified using Fura-2. Scale bar shown as 10 µm. (**b**) Ca^2+^ flux over baseline (no shear) over time as shear stress is ramped from 0 to 1 dynes/cm^2^. Ca^2+^ flux within neutrophil (fold over time 0) is reported as mean +/− SEM over 7 fields of view with >10 cells per experiment (*n* = 3, significance between 54pN and 33/12pN from 24–48 s, *p* < 0.01). (**c**) Percent of total neutrophils that exhibit flux in Ca^2+^ over time, defined as >3-fold increase over baseline, when sheared over TGT functionalized with HA-inducing β_2_-integrin antibody (CBR LFA1/2). (**d**) Histogram of the amount of Ca^2+^ flux over baseline that neutrophils exhibit when sheared over TGT functionalized with HA-inducing β_2_-integrin antibodies (binned every 0.2 MFI/MFI_baseline_). Data are reported as mean +/− SEM over 7 fields of view with >10 cells per experiment (*n* = 3).

**Figure 3 cells-11-02822-f003:**
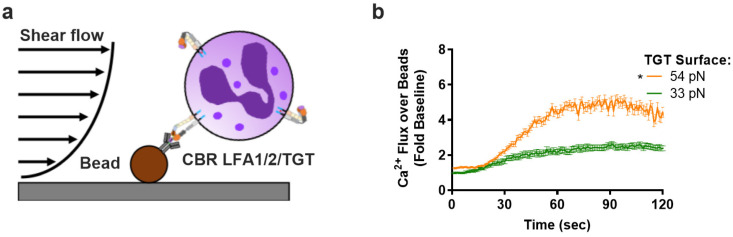
**Neutrophil Ca^2+^ flux increases with****β_2_-integrin****bond tension at equivalent bond number.** (**a**) Schematic showing neutrophil attachment to beads functionalized with TGT/HA β_2_-integrin-inducing antibody (CBR LFA1/2) under shear flow. The cells are allowed to pivot over the bead, and only cells that resist the shear flow and remain bound for 2 min are quantified. (**b**) Ca^2+^ flux of neutrophils that have pivoted over the functionalized beads. Data are reported as mean +/− SEM over 5 fields of view with >3 cells per experiment (*n* = 3, * significance between 54pN and 33pN starting at 40 s).

**Figure 4 cells-11-02822-f004:**
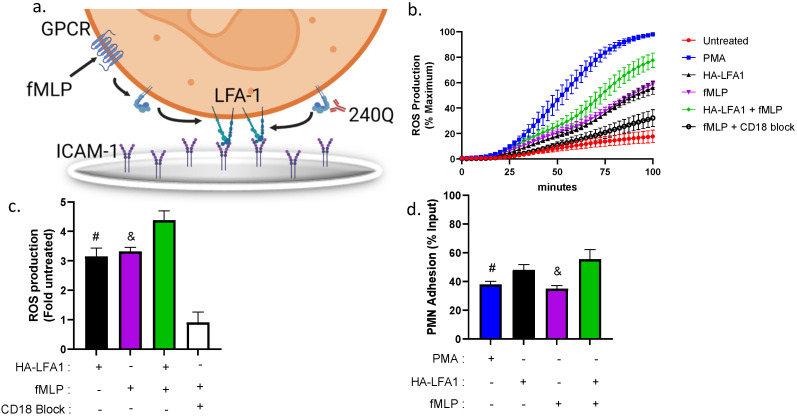
**Mechanoregulation of superoxide production in neutrophils adherent to ICAM****-1.** Neutrophils were allowed to adhere to the substrate of 96 well plates coated with ICAM-1 in the presence of 240Q, fMLP, or both, and the dynamics of intracellular reactive oxygen species (ROS) were measured in a population of shear-mixed neutrophils by detection of dihydrorhodamine (DHR) in a fluorescence plate reader. (**a**) Schematic of cooperative signaling of neutrophil ROS production by GPCR via fMLP and activation of HA CD18 by 240Q binding to ICAM-1. (**b**) Real-time kinetics of ROS production in DHR buffer for neutrophils shear mixed every 2.5 min and stimulated with PMA or 240Q in the presence and absence of fMLP or blocking of CD18 with IB4. (**c**) Histogram of peak ROS production for the indicated conditions expressed as fold increase from the baseline level registered in untreated neutrophils. All data are represented as mean +/− SEM (*n* = 5, # denote significance comparing HA LFA-1 vs. HA LFA-1 + fMLP, *p* < 0.05). & denote significance fMLP vs. HA LFA-1 + fMLP, *p* < 0.01. All conditions significantly greater than CD18 block condition, *p* < 0.01. (**d**) Neutrophil adhesion measured as fraction of total input remaining at the end of the time course based upon Hoechst fluorescence. All data are represented as mean +/− SEM (*n* = 5, # significance comparing PMA vs. HA LFA-1 + fMLP, *p* < 0.01. & denote significance comparing fMLP vs. HA LFA-1 + fMLP, *p* < 0.01).

**Figure 5 cells-11-02822-f005:**
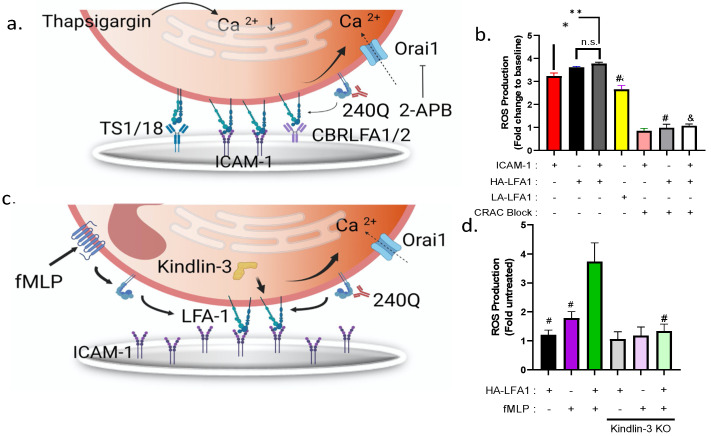
**High-affinity****β****_2_****-****integrin potentiates superoxide production via signaling of kindlin-3-mediated Ca^2+^ influx.** (**a**) Schematic of ROS activity in freshly isolated neutrophils treated with thapsigargin to deplete cytosolic Ca^2+^ and stimulated by addition of buffer containing 1 mM Ca^2+^ and, where denoted, 240Q to induce HA β_2_-integrin before addition to well substrates coated with allosteric antibody to induce HA β_2_-integrin (CBRLFA1/2), LA β_2_-integrin (TS1/18), or ICAM-1 + 240Q as denoted. CRAC channel activation is stimulated by addition of 1.5 mM CaCl_2_ and blocked in the presence of 2-APB. (**b**) Plot of the maximum increase in ROS activity normalized by baseline level of unstimulated cells attached to ICAM-1 for the indicated conditions over 100 min of shear mixing. All data are represented as mean +/− SEM (*n* = 3, * and ** denotes significance compared to ICAM-1 alone *p* < 0.05 and *p* < 0.01 respectivelyt. # denotes significance compared to HA β_2_-integrin alone. & denotes significance compared to ICAM-1, *p* < 0.01). (**c**) Schematic of dHL-60 neutrophils co-stimulated via fMLP and allosteric activation of HA-β_2_-integrin by addition of 240Q to mediate Ca^2+^ entry dependent upon kindlin-3 and Orai1. (**d**) Histogram of maximum increase in ROS production in WT and siRNA to knockdown kindlin-3 activity, normalized by untreated dHL60. All data are represented as mean +/− SEM (*n* = 3, # significance to HA-LFA-1+ fMLP condition, *p* < 0.01).

**Figure 6 cells-11-02822-f006:**
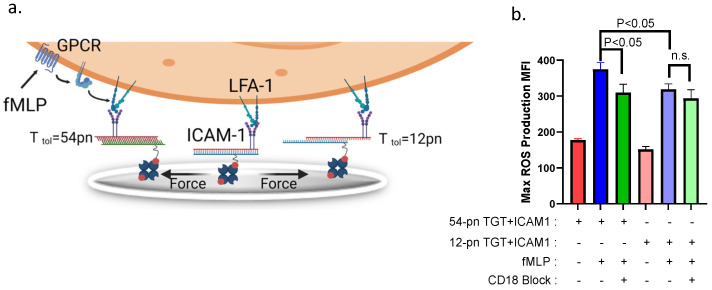
**Bond tension acting on β_2_****-integrins modulates superoxide production in neutrophils.** (**a**) Schematic of neutrophils bound to 12pN or 54pN TGT presenting ICAM-1 in presence and absence of HA β_2_-integrin activated with fMLP. Shear force produced by oscillatory mixing greater than the TGT tolerance results in rupture. (**b**) Max ROS production for neutrophils bound to 54pN or 12pN TGT + ICAM-1 in the presence or absence of fMLP or CD18 blocking antibody IB4. All data are represented as mean +/− SEM, *n* = 3, bars indicate significance between mean values. Note that 12pN TGT + ICAM-1 did not register a significant increase compared to fMLP stimulation in absence of IB4.

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
