# Peer review of "β2-Integrin Adhesive Bond Tension under Shear Stress Modulates Cytosolic Calcium Flux and Neutrophil Inflammatory Response"

_cells, 2022, doi:10.3390/cells11182822_

Round 1

Reviewer 1 Report

The study by Morikis and colleagues extends their recent work on the mechanotransduction events that are involved in the transition of neutrophils from nascent arrest to shape change (flattening) and polarization. Importantly, these processes occur and are regulated by forces on the neutrophil due to fluid shear stress, and this study uses an innovative approach by applying tension gauge tethers to generate new knowledge at the level of single integrins. They demonstrate that a threshold force of ~33pN (or at least 16.5pN considering bivalence) on activated LFA-1 is necessary to trigger calcium influx and downstream adhesive and functional activity of neutrophils. The study is generally well done, using techniques and analyses well established within this group. I have the following critiques and questions that should be addressed:

1. In probing the threshold tension on LFA-1 to mediate downstream calcium influx and adhesion strengthening, anti-LFA-1 antibodies are used to engage LFA-1 in all experiments presented. Did the authors analyze or confirm threshold tension on LFA-1 for these outcomes when the physiologic ligand ICAM-1 is conjugated to the TGT (as was done for ROS production under static conditions)? Otherwise, the authors are making the assumption that HA-specific antibody-bound LFA-1 transduces a mechanosignal in the same way that ICAM-1-bound LFA-1 does – that assumption should be stated explicitly in the absence of data using ICAM-1/TGT.

2. The ROS assays are performed under static conditions (this should be explicitly stated in the Results section), it seems, and so the exertion of tension on LFA-1 is occurring via a different modality than in adhesion/signaling assays under flow. The authors should discuss this and place it in the context of observing similar threshold tensions for LFA-1 signaling under the different conditions.

3. In the Discussion, the authors consider the case of dimeric bonds between antibodies and LFA-1, pointing out that for example a 54pN TGT may rupture when bound to 2 LFA-1 receptors that exceed 27pN threshold tension. This consideration does not detract from the findings of the study at all, but should be mentioned when the TGT system is first being described (end of Introduction or in the Results section).

4. In the first paragraph of Result section 3.1, it states “…anti-LFA-1 allosteric inducing antibodies that upon binding to the integrin’s I-domain stabilize a HA (240Q or CBRLFA-1/2)…” However, CBRLFA-1/2 actually binds to the leg of CD18 and thereby induces HA. This should be revised and it would be helpful to include references for these specific antibodies here.

Reviewer 2 Report

The manuscript titled «β2-integrin adhesive bond tension under shear stress modulates cytosolic calcium flux and neutrophil inflammatory response» contain new interesting data about role of LFA-1 in regulation cytosolic calcium flux and reactive oxygen species production using tension gauge tethers captured to different affinity allosteric anti-LFA-1. Data obtained on good numbers of samples, using donors neutrophils and differenciated HL-60. Statistical analysis carried out with adequate methods. Considering that oxidants generated by neutrophils can damage almost any biopolymer structures, it is possible that the authors should inspire readers with confidence that the tension sensors used in the work are not vulnerable to reactive oxygen species generated by cells. This manuscript strongly appropriate for publication in Cells.

Author Response

Please see attachment,

Reviewer 3 Report

Dr. Morikis, Dr. Simon, and co-authors presented an impressive study showing the LFA-1 biomechanical signaling to trigger calcium flux. My concerns are listed below:

1.            CBRLFA-1/2 does not bind to beta2 integrin α I-domain or β I-like domain but β IEGF-3 domain, as shown in Chen et al. PNAS 2012. Yes, it induces the HA beta2 integrin. But not only LFA-1. So other beta2 integrins should be considered. And should not be said as LFA-1 in the text and figures. Even Mac-1 blocking antibody was used in some assays, it will only block ICAM-1 binding but not antibody binding.

2.            As mentioned in the text, 3 antibodies (240Q, CBRLFA-1/2, and TS1/18) have been used in the assay. So what is the antibody for different figures? You always say HA LFA-1.

3.            Where does the green in Figure 1c come from? The schematic of Figure 1a does not show any fluorescence. And there is no explanation about the green signal in the figure legend. How is the LFA-1 density quantified? It is not clear in the method and the figure legend. If you use the fluorescence to quantify, you need a standard curve.

4.            Scale bars are missing. Microscopy images are too small to assess. Maybe larger images are better to show more details.

5.            Does TS 1/18 TGT have neutrophil adhered? Please show data like Figure 1b.

6.            The pulling force for TS1/18 and CBRLFA-1/2 might be in a different direction since CBRLFA-1/2 does not bind to beta2 integrin α I-domain or β I-like domain but β IEGF-3 domain. Thus CBRLFA-1/2 TGT might provide a horizontal force to extended integrin molecules. The induction of cluster density increase might be caused by the horizontal force, regardless of the affinity state of beta2 integrin.

7.            I am confused by Figures 1e and f. 1e is a “Ca2+ reporter Rhod-2”. So I think it is similar to figure 2. And figure 1e is too small to see the sharp changes you mentioned in the text. Are they epifluorescence or TIRF images?

8.            I was surprised to see that under shear stress from 0-1 dyn/cm2, there is a big difference in Ca2+ between 54 pN and 33 pN groups in Figure 2. Because you have shown that at 1 dyn/cm2, most cells (70-90%) are remain adhered in the 33pN group. So 1 dyn/cm2 should generate ~33pN force on integrin bonds, regardless of TGT substrate, even in the 54 pN substrate. Similar to Figure 1c, which uses 1 dyn/cm2 as well. Can you explain why?

9.            Why did you use beads in Figure 3? Any difference from the substrate coating?

10.          For Figure 4, is there any plates coated with antibodies as mentioned in the text (“coated with either ICAM-1 or allosteric inducing antibody”)? Or only coated with ICAM-1, but some groups added 240Q antibody. If you did antibody coating in Figure 5, please do not mention it in section 3.3, which only discusses Figure 4.

11.          I was confused by Figure 4d. Where is the shear come from for a 96-well plate? And why do you count the cell number?

12.          TS1/18 and IB4 treated ROS assay are mentioned but not cited. Which figures? BTW, as I remember, TS1/18 is also a blocking antibody similar to IB4 but does not stabilize the LA state of beta2 integrins, which you claim in the paper. If you are correct, please cite the paper showing the structure evidence.

13.          Figure 5b, missing an ICAM-1 only negative control. Or was it used as the “untreated” group? What is the untreated group? Have ICAM-1 or not? Also, it is surprising to see that LA-LFA1 (I think you use TS1/18) is ~2.5 fold compared to untreated. How to explain?

14.          Isotype controls are missing when you want to see antibody effect, either blocking, inhibiting, or activating.

15.          Figure 5d, why does HA-LFA-1 drop back to ~1 fold to untreated? It is ~3.5 fold in Figure 5b, and ~3 folds in Figure 4c. This is so confusing.

16.          Figure 6b, no significant changes was shown. It should show some statistical significance.

Round 2

Reviewer 1 Report

Thank you to the authors for addressing all of my comments and questions.

Author Response

Thank you for your suggestions. We appreciate the feedback.

Reviewer 3 Report

Previous point 1: The response does not describe what changes have been made. I cannot go through the whole manuscript and compare it to the old version to find out what changes you have made.

And also, most important, you always said LFA-1 in the paper, but your experiments are focused on the beta2 chain. You can not exclude other beta2 integrins when you describe them. Only using LFA-1 in the text is misleading.

Previous point 2: what is the antibody used for 54, 33, and 12 pN TGT? Still not clear. CBR LFA1/2 or TS1/18?

Previous point 3: Fig. 1a, mAb24 binds to beta2 I-like domain. And the CBR LFA1/2 (allosteric) binds to the star position. So the schematic is wrong.

Fig. 1d, sorry that I am confused. It is great that you did a standard curve, as I suggested. But in Fig. 1d, you show the mAb24 cluster number but not the site density. Why don’t you use the site density?

Previous point 5: if you don’t want to show the graph data, you also need to mention it in the text. Did you? Where?

Previous point 6: you said, “our data supports the conclusion that mechanical tension above a threshold level effectively controls the rate and extent of Ca2+ signaling that modulates subsequent inflammatory function dependent on hydrodynamic stress”. This is not correct. Your data approved that mechanical tension on HA-beta2 integrins but not LA-beta2 integrins above a threshold level effectively controls the rate and extent of Ca2+ signaling that modulates subsequent inflammatory function dependent on hydrodynamic stress.

And I still believe that the pulling geometry is more important. For TS1/18 TGT, integrins are bent-low affinity, and the pulling is from the head α I-domain. And for CBR LFA1/2, integrins are extended-high affinity, and the pulling is from the knee β IEGF-3 domain.

To exclude a pulling geometry issue, you can use Mn2+ to induce HA integrins and use TS1/18 TGT under different pulling forces to test. ICAM-1 binding you mentioned should not be considered because LA integrins will not bind ICAM-1. The control should be LA integrins bound to TS1/18 TGT. I think you did this control in some of your experiments.

Previous point 8: I understand what you mean. My question is that under a shear stress of 1 dyn/cm2, forces most cells (~80% according to Fig. 1b) born should be aournd or less than 33pN, regardless of 33pN or 54pN TGT substrate. So how did you get such big differences in Ca2+? 20% of cells that bear more than 33pN force can generate ~2-3 fold difference. Are there any factors that should be considered or discussed?

Previous point 10: did you make any changes in the text to clarify?

Previous point 11: you answered how you did the cell count, but my question is why. If you want to use the cell count to normalize the data from Figures 4b and c, did you do the normalization, and how? And looks that the groups are not consistent in the 3 figures. Many controls are missing.

Previous point 12: thank you for providing the reference. You still do not show the TS1/18 ROS data, which you mentioned in line 399. Figure 4 only has IB4 data.

Previous point 13: then I suggest you normalize the ROS by cell adhesion number. Are the HA-LFA-1 and LA-LFA-1 have the same number of cells adhered?

Previous point 15: do you mean that in figure 5b, the HA-LFA-1 means the CBR LFA1/2 coating, and in Figure 5d, the HA-LFA1 means 240Q added? If so, please label them separately. Using the same label is confusing.

 Minor points: please check the typos in the text.
